# Adiponectin in the mammalian host influences ticks' acquisition of the Lyme disease pathogen *Borrelia*

Xiaotian Tang[1]*, Yongguo Cao[2], Carmen J. Booth[3], Gunjan Arora[1¤], Yingjun Cui[1], Jaqueline Matias[1], Erol Fikrig[1]

1 Section of Infectious Diseases, Department of Internal Medicine, School of Medicine, Yale University, New Haven, Connecticut, United States of America, 2 College of Veterinary Medicine, Jilin University, Changchun, China, 3 Department of Comparative Medicine, Yale School of Medicine, New Haven, Connecticut, United States of America

¤ Current address: National Heart Lung & Blood Institute, Bethesda, Maryland, United States of America
* xiaotian.tang@yale.edu

**Data Availability Statement:** All data are available in the main text or the supplementary materials. All the RNA-seq data sets were used in this study are available in the supplementary data or the National

## Abstract

Arthropod-borne pathogens cause some of the most important human and animal infectious diseases. Many vectors acquire or transmit pathogens through the process of blood feeding. Here, we report adiponectin, the most abundant adipocyte-derived hormone circulating in human blood, directly or indirectly inhibits acquisition of the Lyme disease agent, *Borrelia burgdorferi*, by *Ixodes scapularis* ticks. Rather than altering tick feeding or spirochete viability, adiponectin or its associated factors induces host histamine release when the tick feeds, which leads to vascular leakage, infiltration of neutrophils and macrophages, and inflammation at the bite site. Consistent with this, adiponectin-deficient mice have diminished pro-inflammatory responses, including interleukin (IL)-12 and IL-1β, following a tick bite, compared with wild-type animals. All these factors mediated by adiponectin or associated factors influence *B. burgdorferi* survival at the tick bite site. These results suggest a host adipocyte-derived hormone modulates pathogen acquisition by a blood-feeding arthropod.

## Introduction

Hematophagous arthropods have evolved the ability to ingest large amounts of blood in a single feeding, in particular, ticks can ingest as much as 100 times their initial weight [1]. Blood contains a variety of nutrients, cytokines/chemokines, microbes, and pathogen-derived molecules. Blood components or their metabolites can directly or indirectly modulate lifespan, reproduction, and the immune and physiological status of arthropods. These factors may influence arthropod susceptibility to the pathogens they can transmit [2]. Indeed, factors in host blood may regulate pathogen colonization, including plasmodium and arboviruses in mosquitoes and spirochetes in ticks. For example, serum iron in human blood inhibits dengue virus infection in mosquitoes by boosting the activity of reactive oxygen species (ROS) in the gut epithelium [3]. A host-derived cytokine in blood, interferon-gamma (IFN-γ), can interact with tick receptor Dome1 and activate the STAT-dependent pathway, expressing the

Center for Biotechnology Information (NCBI) Sequence Read Archive (GSE233658).

**Funding:** This work was supported by grants from the National Institutes of Health (AI126033 to EF; AI138949 to EF) and the Steven and Alexandra Cohen Foundation (to EF). This research was also supported by the Howard Hughes Medical Institute Emerging Pathogens Initiative (to EF). The funders had no role in study design, data collection and analysis, decision to publish, or preparation of the manuscript.

**Competing interests:** The authors have declared that no competing interests exist.

**Abbreviations:** GO, Gene Ontology; HBP, histamine-binding protein; HE, hematoxylin and eosin; IACUC, Institutional Animal Care and Use Committee; KEGG, Kyoto Encyclopedia of Genes and Genomes; KO, knock out; qPCR, quantitative real-time PCR; ROS, reactive oxygen species; SPF, specific pathogen-free; tHRF, tick histamine release factor; WT, wild-type; YCGA, Yale Centre for Genome Analysis.

borreliacidal peptide and limiting *Borrelia burgdorferi*, persistence within *Ixodes scapularis* [4,5]. The pathogens, in turn, have evolved to neutralize the detrimental effect from host-derived factors within arthropods. *B. burgdorferi* expresses the outer surface proteins OspA and OspB, which protect spirochetes from harmful components in host blood, including antibodies and complement, and enable them to persist in the tick gut [6–8]. Therefore, expanding our understanding of the complex interactions among vector, pathogen, and host can guide novel approaches for the control of a variety of arthropod-borne diseases.

Adiponectin is an adipocyte-derived hormone and is the most abundant adipokine circulating in human blood, with a normal circulating level of 3 to 30 μg/mL [9]. Adiponectin has varied functions in multiple organs, including the liver, heart, and kidney. Available evidence suggests that adiponectin is involved in a variety of biological processes through binding adiponectin receptors, including lipid metabolism, energy regulation, and insulin sensitivity [10]. In addition to metabolism modulation, adiponectin also has key roles in immunity, exhibiting both pro- and anti-inflammatory effects [11]. For example, adiponectin stimulates the secretion of chemotactic factors and increases interleukin (IL)-6 production in human adipocytes. Furthermore, its high serum level was positively associated with inflammation severity and pathological progression in rheumatoid arthritis [11]. Adiponectin can also function as an activity modulator of mast cells [12], a type of white blood cell found in the subcutaneous and connective tissues in mice. The abundance in blood and variable function of adiponectin suggest that adiponectin could be involved in the regulation of host responses, arthropod physiology, and/or pathogen infection during vector feeding. Our recent study indicated a tick adiponectin receptor-like protein regulates phospholipid metabolism in tick gut, which is associated with colonization by the Lyme disease agent *B. burgdorferi* [13]. Moreover, an adiponectin-homolog in tick, complement C1q-like protein 3, alters infectivity of *B. burgdorferi* by modulating host IFN-γ production [14]. In the present study, we investigated whether host serum adiponectin or its associated factors influence *B. burgdorferi* acquisition or colonization by *I. scapularis*, since host blood is taken into the tick during a blood meal, and this is the primary method by which ticks acquire *B. burgdorferi*.

## Results

### Adiponectin deficiency affects *Borrelia* acquisition by ticks upon blood feeding

Since spirochetes enter ticks during a blood meal, we investigated whether host serum adiponectin influences *B. burgdorferi* acquisition or colonization by *I. scapularis*. B6;129-*Adipoq*<sup>tm1-Chan</sup>/J mice deficient in *adiponectin* null or knock out (KO) and their wild-type (WT) littermates (*n* = 5 each) were first needle inoculated with *B. burgdorferi*. No significant difference in *B. burgdorferi* burden was observed between the 2 groups of mice (Fig 1A) ($P > 0.05$). Three infected mice with a similar *B. burgdorferi* burden were then subjected to being fed upon by uninfected *I. scapularis* nymphs (Fig 1B). The number of *B. burgdorferi* in the tick gut was then evaluated by quantitative real-time PCR (qPCR). At 48 h, the spirochete levels in the ticks feeding on KO mice were significantly higher ($p < 0.01$) compared to the ticks feeding on WT mice (Fig 1C). This suggests murine adiponectin may affect *B. burgdorferi* acquisition by ticks upon blood feeding. We further investigated whether adiponectin modulates tick feeding and therefore indirectly influences *B. burgdorferi* acquisition. The attachment and engorgement weights of nymphs feeding on infected WT or KO mice were comparable (Fig 1D and 1E) and suggest adiponectin does not alter the ability of ticks to take a blood meal. As adiponectin encodes a C1q-like globular domain, and numerous C1q domain-containing proteins have high affinity for microbes [14–17], we then examined whether adiponectin directly

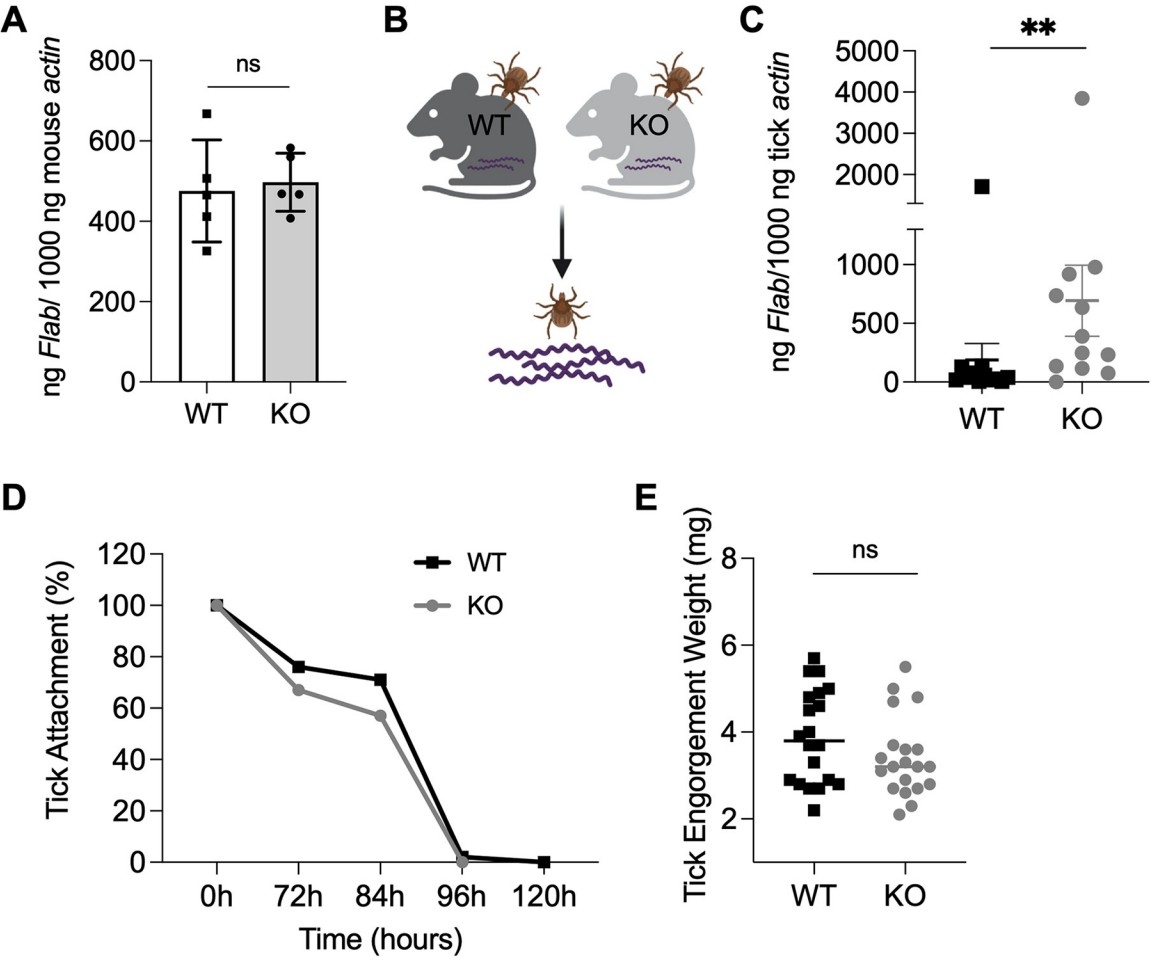

**Fig 1. Adiponectin affects *B. burgdorferi* acquisition by ticks upon blood feeding.** (A) The *B. burgdorferi* burden in WT (*n* = 5) and KO mice (*n* = 5) after 14 days infection. The mice were needle inoculated with *B. burgdorferi*. (B) Pathogen-free *I. scapularis* nymphs were fed on *B. burgdorferi*-infected WT (*n* = 3) and KO mice (*n* = 3) for 48 h and then *B. burgdorferi flaB* levels in guts were assessed. (C) *B. burgdorferi* burden in the tick gut after feeding on WT and KO mice for 48 h. *B. burgdorferi* titers in the ticks feeding on KO mice were significantly higher compared to the ticks feeding on WT mice. (D) *B. burgdorferi* infected WT (*n* = 3) and KO mice (*n* = 3) were challenged with ticks and assessed for tick detachment. The percentage of ticks remaining attached on mice at a given time point was recorded. Three replicates were included, and the average values were presented. (E) The engorgement weights of nymphs feeding on WT (*n* = 3) and KO mice (*n* = 3). For all the data, each dot represents 1 biological replicate. Statistical significance was assessed using a nonparametric Mann–Whitney test (**$p < 0.01$; ns, $p > 0.05$). Data underlying this figure can be found in S1 Data. We would like to acknowledge that figures were created using BioRender (https://www.biorender.com/) with permission. KO, knock out; WT, wild-type.

interacts with *B. burgdorferi*. Flow cytometry showed adiponectin does not directly bind *B. burgdorferi* (Fig 2A). In addition, adiponectin had no effect on *B. burgdorferi* viability as assessed by the BacTiter-Glo microbial cell viability assay (Fig 2B). These results suggest adiponectin may influence *B. burgdorferi* acquisition by altering host or tick responses.

## Host adiponectin significantly increases tick histamine-binding protein expression

We examined whether incoming blood adiponectin or associated factors influence tick physiology or alters the local host environment at the tick bite site. We used RNA-sequencing to compare the transcriptomes of ticks feeding on WT (*n* = 3) and KO mice (*n* = 3) (Fig 2C). Upon blood meal feeding, 221 genes were significantly differentially expressed in ticks feeding

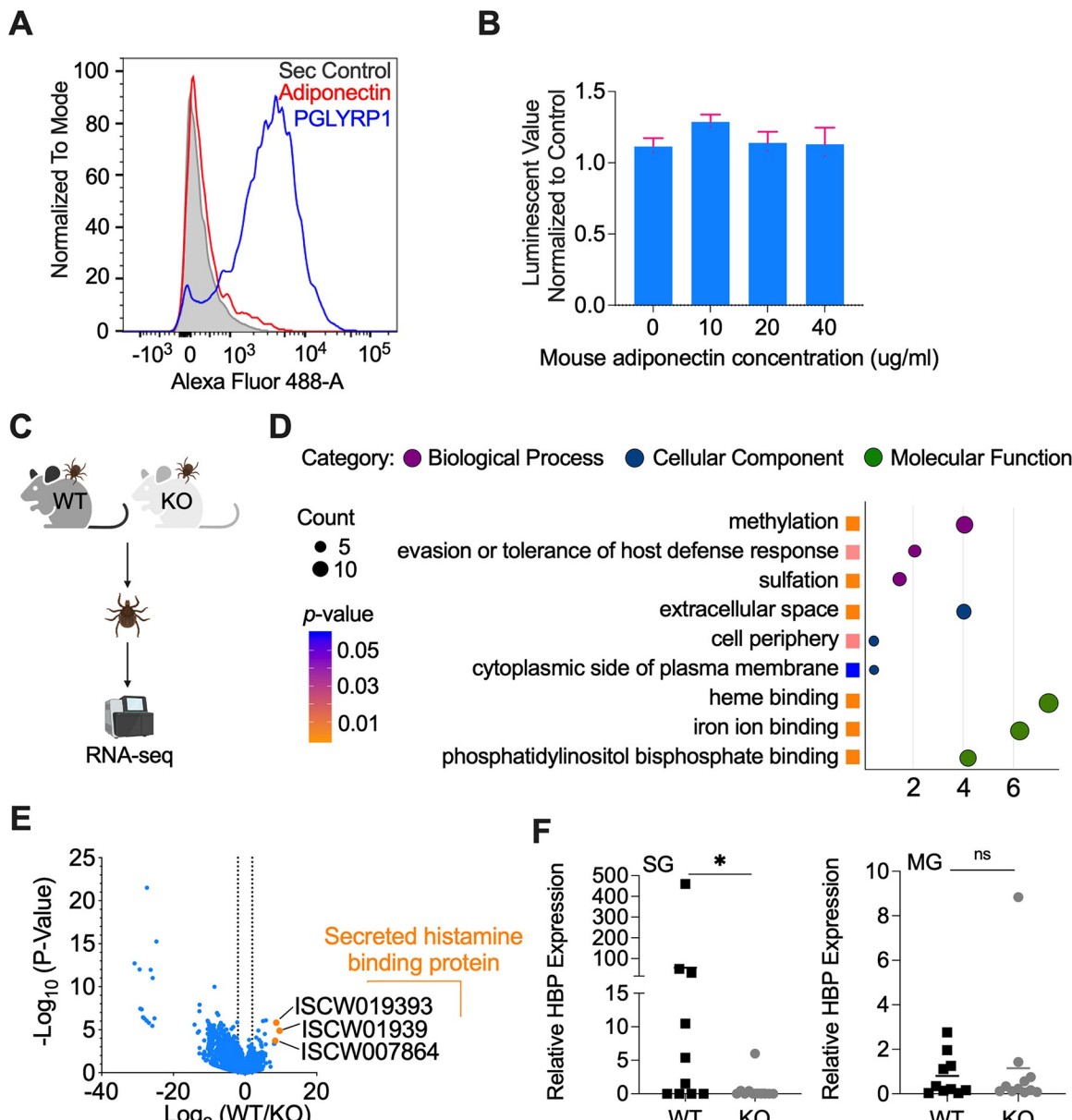

**Fig 2. Host adiponectin significantly increases tick HBP expression.** (A) No interaction of adiponectin with *B. burgdorferi* was identified, as analyzed by flow cytometry. PGLYRP1 was used as positive control. The background of Alexa Fluor 488-His antibody alone with *B. burgdorferi* is shown in gray. (B) Mouse adiponectin has no effect on *B. burgdorferi* viability as determined by BacTiter-Glo assay. (C) RNA-seq of ticks feeding on WT ($n = 3$) and KO mice ($n = 3$). The ticks that fed on the same mouse were pooled for RNA extraction. (D) GO enrichment analysis of transcriptome data from the ticks feeding on WT ($n = 3$) and KO mice ($n = 3$). The second level GO terms were shown in the plot and enrichment analysis was performed using the functional annotation tool DAVID. (E) Volcano plot of differentially expressed genes between the ticks feeding on WT ($n = 3$) and KO mice ($n = 3$). The Top 3 genes were highlighted by orange color. The gene names can be found in S1 Table. (F) qPCR validation of HBP gene expression in SGs and MG of the ticks feeding on WT ($n = 3$) and KO mice ($n = 3$). The ticks feeding on different mice were collected for analysis. For all the data, each dot represents one biological replicate. Statistical significance was assessed using a nonparametric Mann–Whitney test (*$p < 0.05$; ns, $p > 0.05$). Data underlying this figure can be found in S1 Data. We would like to acknowledge that figures were created using BioRender (https://www.biorender.com/) with permission. GO, Gene Ontology; HBP, histamine-binding protein; KO, knock out; MG, midgut; PGLYRP1, peptidoglycan recognition protein 1; qPCR, quantitative real-time PCR; SG, salivary gland; WT, wild-type.

on WT mice when compared to that feeding on KO mice (S1 Table). Based on Gene Ontology (GO) functional classification analyses, the most differentially expressed genes in the biological process category were involved in methylation (GO: 0032259) and evasion of host defense responses (GO: 0042783) (Fig 2D). Interestingly, the top 3 up-regulated genes (ISCW019392, ISCW019393, and ISCW007864) were all secreted histamine-binding proteins (HBPs) (Fig 2E), which fell into the GO category of evasion of the host defense response (GO: 0042783) and were validated in tick salivary gland by qPCR (Fig 2F). We were mostly interested in the HBPs because HBPs can bind histamine, which is secreted by host immune cells (e.g., mast cells and basophils), to the advantage of tick feeding. Importantly, histamine-related proteins may also be involved in *B. burgdorferi* infection. For example, blocking tick histamine release factor (tHRF) by the passive transfer of tHRF antiserum to mice can diminish *B. burgdorferi* infectivity [18].

## KO mice secrete less histamine, which causes less vascular leakage at the tick bite site

Since HBPs were most differentially expressed between ticks feeding on WT and KO mice, and adiponectin can function as a mast cell activity modulator [12], it is possible WT and KO mice may secrete distinct levels of histamine during tick feeding. To test this hypothesis, we evaluated the histamine level during a tick bite and found that WT mice ($n = 10$) have significantly higher levels of histamine compared to KO mice ($n = 10$) after 2 days (Fig 3A). Histamine has been shown to strongly increase vascular leakage [19,20]. Therefore, we further assessed murine vascular leakage during a tick bite by intravenously injecting Evans blue into the WT ($n = 6$) and KO mice ($n = 6$). The results indicated WT mice have more vascular leakage than KO mice at the tick bite site (Fig 3B), and the quantification of Evans blue further confirmed the vascular leakage difference between WT ($n = 6$) and KO mice ($n = 6$) (Fig 3B). Therefore, KO mice secrete less histamine during a tick bite, which causes less vascular leakage at the tick bite site.

## KO mice exhibited less immune cellular infiltration and inflammation at the tick bite site

There was a difference in histamine-mediated vascular leakage between these 2 types of mice, and vascular permeability allows for platelets and immune cells to reach the site of tissue damage [21]. We therefore investigated whether during tick feeding, histamine increases blood vessel permeability to allow immune cells to access the tick bite site. We focused on immune cells, which infiltrate the tick bite site, including neutrophils, dendritic cells, macrophages, basophils, and mast cells [22]. Through flow cytometry, we found that more neutrophils ($CD11b^+Ly6G^+/CD45^+$) and macrophages ($CD11b^+CD11c^-/CD45^+Ly6G^-$) were recruited to the tick bite site of WT mice ($n = 4$) compared to KO mice ($n = 4$) ($p < 0.05$) (Fig 3C), while dendritic cells ($CD11b^-CD11c^+/CD45^+Ly6G^-$) and mast cells ($c\text{-kit}^+CD49b^+CD200R3^+$) populations were similar ($p > 0.05$). Basophils ($c\text{-kit}^-CD49b^+CD200R3^+$), as expected in mice, were few and not counted in this study. Histamine causes inflammation by promoting vascular leakage [20,23], and there was a difference in inflammatory cells recruitment between the 2 types of mice.

To compare the pathologic changes at the tick bite site, semiquantitative histopathologic analysis was performed on ears of WT ($n = 5$) and KO ($n = 4$) mice euthanized 96 h post tick feeding. WT mice had a total of 13 tick bite/inflammatory foci (10 ears total) and KO mice had a total of 11 tick bite/inflammatory foci (8 ears total). Each ear was evaluated for the presence and number of tick bite/inflammatory foci. Then, each focus was further evaluated and scored

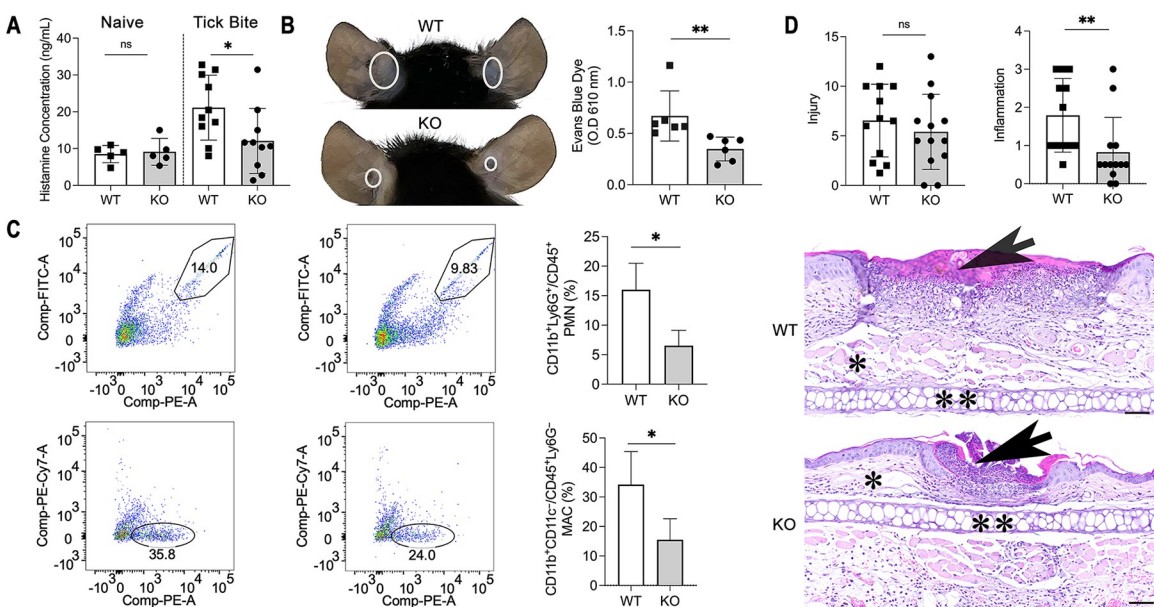

**Fig 3. Adiponectin-deficient mice exhibit less inflammation at the tick bite site.** (A) Histamine concentration in WT ($n = 5$) and KO naïve mice ($n = 5$), or after tick bite ($n = 10$). (B) Injection of Evans blue during tick feeding. The white circles indicated WT ($n = 6$) mice have more vascular leakage than KO mice ($n = 6$) at the tick bite site. Quantification of Evans blue leakage at the tick bite site of WT ($n = 6$) and KO mice ($n = 6$). (C) The CD11b$^+$Ly6G$^+$/CD45$^+$ PMN population and the percentage of PMNs in the total CD45$^+$ leukocyte cell population in WT ($n = 4$) and KO mice ($n = 4$) after tick bite. The CD11b$^+$CD11c$^-$/CD45$^+$Ly6G$^-$ MAC population and the percentage of MACs in the total CD45$^+$ leukocyte cell population in WT ($n = 4$) and KO mice ($n = 4$) after tick bite. (D) Semiquantitative histopathologic scoring of tick bite/inflammation sites show there is no significant difference in the severity of injury (Injury) but there is an overall increase in the degree of inflammation (Inflammation) in WT mice ($n = 5$) compared to adiponectin KO mice ($n = 4$). Representative HE-stained sections of WT and adiponectin KO tick bite lesions (arrows). Scale bars = 50 μm, * = subcutis, and ** = ear cartilage. For all the data, statistical significance was assessed using a nonparametric Mann–Whitney test (*$p < 0.05$; **$p < 0.01$; ns, $p > 0.05$). Data underlying this figure can be found in S1 Data. HE, hematoxylin and eosin; KO, knock out; MAC, macrophage; PMN, polymorphonuclear neutrophil; WT, wild-type.

for the presence and severity of the following parameters: edema, hemorrhage, fibrin/necrosis, and inflammation. While there was no significant difference in the average for WT (6.54) and KO (5.40) in overall severity of injury per tick bite (sum of individual parameters) (Fig 3D), there was a significant difference in the severity of inflammation for tick bites between the 2 groups WT mice (average 1.79) compared to KO mice (0.83) ($p < 0.01$) (Fig 3D).

ROS are also key signaling molecules that play an important role in the progression of inflammatory disorders, and adiponectin activates mast cells to produce ROS in vitro [12]. We therefore further examined the ROS level at the tick bite site. No significant difference of ROS level was observed between WT ($n = 4$) and KO mice ($n = 4$) during the tick bite (S1 Fig).

## KO mice showed an attenuated inflammation signature at the tick bite site

We also assessed the production of cytokines and chemokines in the serum or at the tick bite site of WT and KO mice. First, upon being bitten by ticks, serum cytokine/chemokine profiles in mice were assessed using a murine cytokine/chemokine array panel (Fig 4A). We found that WT mice ($n = 5$) have a higher production of 2 cytokines, IL-12p40 and IL-13, compared to KO mice ($n = 5$) during tick feeding (Figs 4A and S2). We then evaluated the expression of immune-related genes at the tick bite site. We selected representative genes of cytokine, chemokine, and inflammation-related receptor, which were listed in our previous study [24]. We find the expression of 2 pro-inflammatory cytokines, IL-1β and IL-12, in WT mice were significantly higher in WT mice ($n = 6$) compared to KO mice ($n = 6$) (Figs 4B and S3). Toll-like

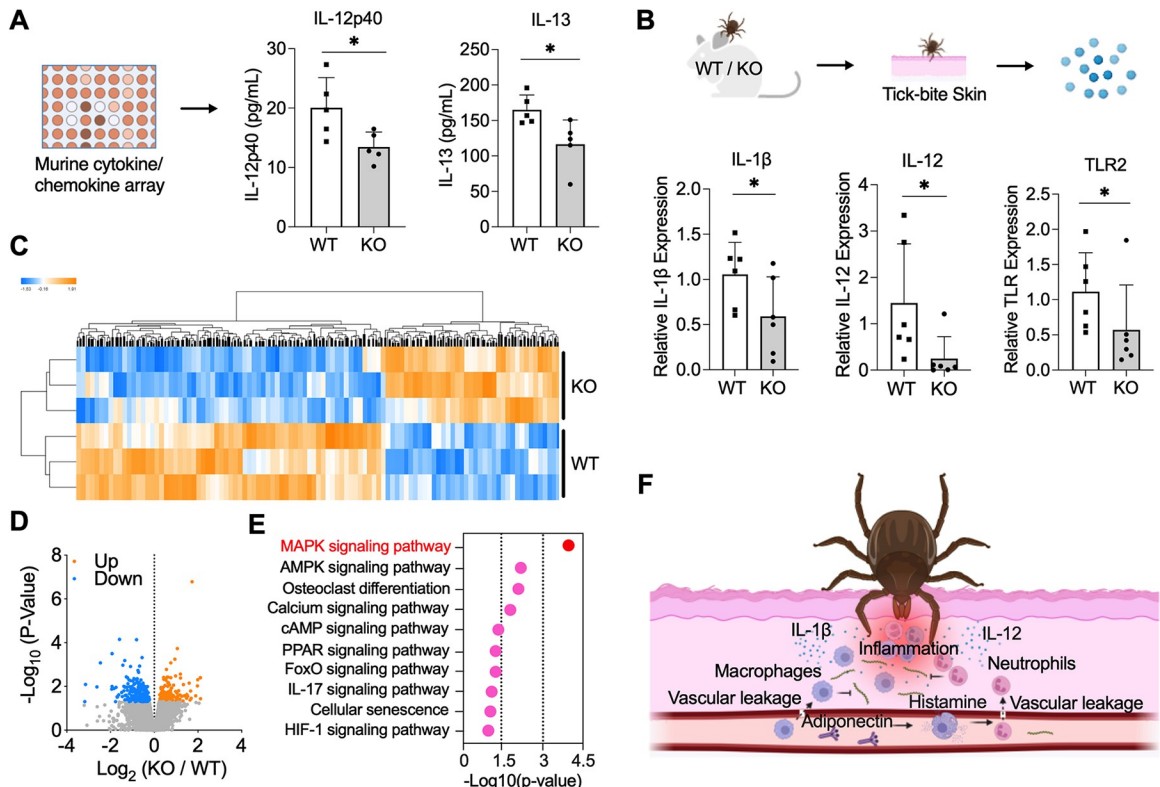

**Fig 4. Adiponectin-deficient mice showed an attenuated inflammation signature at the tick bite site.** (A) Serum cytokines and chemokines production in WT (*n* = 5) and KO mice (*n* = 5) after tick feeding. WT mice have higher production of IL-12p40 and IL-13. (B) Gene expression of cytokines and chemokines at the tick bite site of WT (*n* = 6) and KO mice (*n* = 6) after tick feeding. WT mice have higher gene expression of IL-1β, IL-12, and TLR2. For all the data, each dot represents 1 biological replicate. Statistical significance was assessed using a nonparametric Mann–Whitney test (\**p* < 0.05). (C) Cluster dendrogram and heatmap of transcriptome data of WT (*n* = 3) and KO murine skin (*n* = 3) during tick bite. (D) Volcano plot of differentially expressed genes. The significant differentially expressed genes were highlighted in blue and orange. (E) Immune signaling pathways identified by KEGG pathway enrichment analysis, which was performed using functional annotation tool DAVID. MAPK signaling pathway is the most enriched pathway. (F) Schematic diagram of the mechanism that adiponectin or associated factors induce inflammation during tick bite, inhibiting acquisition of the Lyme disease agent. Data underlying this figure can be found in S1 Data. We would like to acknowledge that figures were created using BioRender (https://www.biorender.com/) with permission. KEGG, Kyoto Encyclopedia of Genes and Genomes; KO, knock out; WT, wild-type.

receptor 2 (TLR2), which is required for production of pro-inflammatory cytokines and chemokines including IL-1β and IL-12, was also significantly expressed in WT mice.

To further understand the global gene expression profile at the tick bite site of WT and KO mice, we utilized RNA-seq to compare the transcriptome of WT and KO murine skin (*n* = 3 each) during tick feeding. A total of 266 genes were significantly down-regulated, and 152 genes were up-regulated in KO mice compared to WT mice (Fig 4C and 4D and S2 Table). In addition to the metabolism pathways such as fatty acid metabolism and adipocytokine signaling pathway, the down-regulated genes were also highly enriched in immune-related pathways based on Kyoto Encyclopedia of Genes and Genomes (KEGG) pathway analysis. MAPK signaling pathway, essential to the production of pro-inflammatory cytokines, is the most enriched pathway (*p* < 0.0001) (Fig 4E). The other pro-inflammatory pathways including calcium signaling, FoxO signaling, IL-17 signaling, and HIF-1 signaling pathways were attenuated in KO mice as characterized by the down-regulation of genes including FBJ osteosarcoma oncogene B (*Fosb*), RAS-related protein 1a (*Rap1a*), amphiregulin (*Areg*), insulin-like growth

factor I receptor (*Igf1r*), CD36 molecule (*Cd36*), chemokine (C-X-C motif) receptor 4 (*Cxcr4*), and more. Taken together, KO mice exhibited lower level of inflammation at the tick bite site.

## Adiponectin or associated factors influence *B. burgdorferi* survival at the tick bite site

Since the differences in immune cells and the level of inflammation at the tick bite site were observed between the WT and KO mice, we further evaluated *B. burgdorferi* survival at the tick bite site by culturing the skin biopsy specimens from WT (*n* = 4) and KO mice (*n* = 4). In total, 19 and 18 tick bite skin biopsies were collected from WT and KO mice, respectively. After culturing for 25 days, 7 (out of 19) and 8 (out of 18) skin samples from tick attachment sites of WT and KO mice were *B. burgdorferi*–positive without contamination, respectively. Within the positive samples, the *B. burgdorferi* survival was further measured using a BacTiter-Glo microbial cell viability assay. Indeed, we found *B. burgdorferi* in the tick bite biopsy from KO mice has higher total viability than in WT mice (*p* < 0.05) (S4A Fig). We further stimulated histamine release in KO mice (*n* = 4) by intradermally injecting anti-dinitrophenyl (DNP) IgE [20]. Through this rescue experiment, we examined whether there was a reduction in differences in *B. burgdorferi* acquisition. There was still a higher *B. burgdorferi* burden in ticks feeding on KO mice, but the difference was not significant (*p* = 0.0767) (S4B Fig). All these data suggest that adiponectin or its associated factors influence *B. burgdorferi* survival at the tick bite site.

## Discussion

Blood feeding is a major method by which arthropods acquire pathogens. Recent studies show host blood components can influence pathogen colonization within vectors by affecting vector gut physiology, competence, and microbiota composition [3,4,25,26]. However, the host-derived factors that influence pathogen acquisition at the interface of the feeding site remain largely unknown. Adiponectin is an abundant adipocyte-derived hormone in human blood and plays important roles in regulation of glucose and lipid metabolism, inflammation, and oxidative stress [27]. Here, we find adiponectin deficiency influences *B. burgdorferi* acquisition by ticks.

We show adiponectin has no direct effect on tick blood feeding and *B. burgdorferi* infectivity. Instead, adiponectin (or associated factors) influences *B. burgdorferi* acquisition by immunoregulation of the local environment at the tick bite site. Adiponectin has been shown to exhibit anti-inflammatory or pro-inflammatory effects depending on conditions [11]. Here, adiponectin exhibits pro-inflammatory effects through histamine-mediated immunoregulation at the tick bite site. Histamine is mainly secreted by basophils in blood and mast cells in tissues. Adipocytokines, such as leptin, have been demonstrated to stimulate mast cell to release histamine in vitro [12]. Further, we find adiponectin (or associated factors) may modulate immune cell activity to release histamine in vivo, since WT mice secrete more histamine compared to KO mice during tick feeding. Although higher histamine levels were observed in WT mice, adiponectin modulates *B. burgdorferi* acquisition without affecting tick feeding. Indeed, *I. scapularis* nymphs can feed well on normal mice even following repeated tick infestations [28,29]. Histamine is a mediator of the itch response and promotes the recruitment of pro-inflammatory cells to injury site. In our study, increased vascular leakage was observed in WT mice. The vascular leakage and/or other associated factors may lead to the additional recruitment of neutrophils and macrophages to the tick bite site, resulting in severe inflammation. We also found that the pro-inflammatory interleukins IL-12 and IL-1β were significantly increased in WT mice compared to KO mice. In particular, IL-12, which is produced

predominantly by dendritic cells, macrophages, and neutrophils [30], was significantly increased in both skin and serum. Furthermore, the expression of TLR2, which promotes immune cell activation leading to tissue inflammation [31], was also significantly increased in WT mice. The RNA-seq data showed that MAPK signaling pathway was most highly enriched, and inflammation could be mediated through TLR2-MAPK signaling.

The levels of blood components may vary substantially among individuals, according to their genetic, physiological, and nutritional status. As examples, adiponectin levels are significantly higher in women than in men [32], and low levels of adiponectin are associated with obesity, type 2 diabetes, and atherosclerosis [33]. The findings in our study suggest that animals or humans with distinct adiponectin level may influence *B. burgdorferi* acquisition. In summary, we have discovered that host serum adiponectin or its associated factors may affect the initial entry of *B. burgdorferi* into ticks. Adiponectin or other factors affect histamine release, which leads to vascular leakage and causes severe inflammation with more immune cell infiltration at the tick bite site—thereby altering *B. burgdorferi* acquisition by ticks (Fig 4F). Our findings demonstrate that a host adipocyte-derived hormone can modulate pathogen acquisition by arthropod vector. This paradigm may also be applied to other microbes transmitted by diverse arthropods.

## Materials and methods

### Ethics statement

We performed all the experiments following the Guidelines for the Care and Use of Laboratory Animals of the NIH. The animal protocols in this study were approved by the Institutional Animal Care and Use Committee (IACUC) at the Yale University School of Medicine. The committee has approved Flagellar and Vector-borne Diseases: Pathogenesis and Protection (Protocol Permit Number: 2023–07941) with an approval period of 1/27/2023 to 12/31/2025. Yale Assurance Number is D16-00146 with an approval period of 5/4/2023-5/31/2027.

### Animals, ticks, spirochetes, and cells

Breeding pairs of adiponectin heterozygous ($Adipo^{+/-}$) B6;129-$Adipoq^{tm1Chan}$/J mice were purchased from the Jackson Laboratory. Adiponectin KO ($Adipo^{-/-}$) mice and their WT ($Adipo^{+/+}$) littermates were generated as previously described [34]. The WT, heterozygous, and KO mice were genotyped by the primers: WT forward primer: 5-TGGATGCTGCCATGTTCCCAT-3; WT reverse primer: 5-CTTGTGTCTGTGTCTAGGCCTT-3; and KO reverse primer: 5-CTCCAGACTGCCTTGGGA-3 (S3 Table). Only 7- to 8-week-old female $Adipo^{+/+}$ or WT and null or KO littermate progeny were used for all experiments. All mice were maintained in a specific pathogen-free (SPF) facility at Yale University. The mice used in this study were all female mice. To obtain *B. burgdorferi*-infected mice, the mice were injected subcutaneously with $1 \times 10^5$ cells/mL *B. burgdorferi*.

SPF *I. scapularis* larvae were obtained from the Oklahoma State University (Stillwater, Oklahoma, United States of America). The larval ticks were fed on SPF mice and allowed to molt to nymphs on an approved IACUC protocol. The number of ticks and mice used for each experiment in this study are described below. The ticks were maintained at 85% relative humidity with a 14 h light and 10 h dark period at 23˚C.

*B. burgdorferi* (strain N40) were grown in Barbour–Stoenner–Kelly H (BSK-H) complete medium (Sigma-Aldrich, #B8291) in a 33˚C setting incubator. The live cell density was determined by dark field microscopy and hemocytometer (INCYTO, #DHC-N01).

## Adiponectin effect on *B. burgdorferi* acquisition

To examine the effect of adiponectin on the acquisition of *B. burgdorferi* by the tick, 10 SPF *I. scapularis* nymphs were placed on each *B. burgdorferi*-infected WT ($n = 3$) and KO mouse ($n = 3$) for feeding. At 48 h, the attached ticks were then collected for gut dissection. The *B. burgdorferi* burden in the tick gut was assessed by quantitative real-time PCR (qPCR). qPCR was performed using iQ SYBR Green Supermix (Bio-Rad, #1725124) with an initial denaturing step of 2 min at 95°C and 45 amplification cycles consisting of 20 s at 95°C followed by 15 s at 60°C, and 30 s at 72°C. The primers for qPCR are listed in S3 Table. The *B. burgdorferi* burden was estimated with a method described in our previous studies [24,35,36]. Specifically, *B. burgdorferi flagellin* gene (*flab*), mouse *actin* gene, and tick *actin* gene primers (S3 Table) were first used to amplify the *flab* and *actin* gene, respectively. The concentration of PCR products was then quantified using NanoDrop 2000/2000c Spectrophotometers (Thermo Scientific, # ND2000CLAPTOP). The standard curve of qPCR Cq value and amplicon amount was derived from a series of known DNA dilutions of each target gene. The *B. burgdorferi* burden was quantified by extrapolation from the standard curve and the data were normalized to mouse or tick actin and reported as ng *flab*/ 1,000 ng *actin*.

## Flow cytometry to test *B. burgdorferi*–adiponectin interaction

*B. burgdorferi* were cultured to a density of approximately $10^6$–$10^7$ cells/mL and harvested at $5,000 \times$ g for 15 min. Cells were washed twice with PBS and then blocked in 1% BSA for 1 h at 4°C. After centrifugation, the pellet was suspended and incubated with recombinant adiponectin protein (SinoBiological, #50636-M08H) at 4°C for 2 h. After co-incubation, spirochetes were washed 3 times with PBS and fixed in 2% PFA. After washing, the spirochetes were probed anti 6X-His monoclonal antibody-conjugated to Alexa Fluor 488 (Thermo Fisher, #MA1-21315-488) and run through BD LSRII (BD Bioscience). The data was then analyzed by FlowJo.

## *B. burgdorferi* viability evaluation

To test whether adiponectin influences *B. burgdorferi* viability, $10^6$ cells/mL *B. burgdorferi* were incubated with $1 \times$ PBS, 10, 20, 40 μg/mL adiponectin for 24 h. The viability of *B. burgdorferi* was then evaluated using BacTiter-Glo Microbial Cell Viability Assay kit (Promega, #G8230).

## RNA-seq of ticks and bioinformatic analyses

Ten *I. scapularis* nymphs were placed on each WT ($n = 3$) and KO mouse ($n = 3$). Then, the ticks were collected for gut and salivary gland dissection. The ticks fed on the same mouse were pooled for RNA extraction. Total RNA was purified using RNeasy Mini Kit (Qiagen, # 74104). The RNA samples were then submitted for library preparation using TruSeq (Illumina, San Diego, California, USA) and sequenced using Illumina HiSeq 2500 by paired-end sequencing at the Yale Centre for Genome Analysis (YCGA). The *I. scapularis* transcript data were downloaded from the VectorBase and indexed using the kallisto index. The reads from the sequencer were pseudo-aligned with the index reference transcriptome using kallisto [37]. The counts generated from 3 biological replicates each treatment were processed by DESeq2 [38] in RStudio. The significant genes were then determined by the *p*-value and fold change ($p < 0.05$ and fold change $\geq 2$). Volcano plot was performed on the genes that were differentially expressed. GO enrichment analysis were conducted using the functional annotation tool DAVID (https://david.ncifcrf.gov/tools.jsp).

## Gene expression evaluation by qPCR

To evaluate expression of target genes in ticks, 10 *I. scapularis* nymphs were placed on each WT ($n = 3$) and KO mice ($n = 3$). The attached *I. scapularis* nymphs were then dissected under the dissecting microscope to get guts and salivary glands. The RNA from dissected guts and salivary glands were purified by Trizol (Invitrogen, #15596–018) following the manufacturer's protocol, and cDNA was synthesized using the iScript cDNA Synthesis Kits (Bio-Rad, #1708891). qPCR was performed as described above. The primers for qPCR are listed in S3 Table. The relative expression of the candidate genes was estimated with the $2^{-\Delta\Delta CT}$ method [39].

## Histamine quantification

To assess the histamine concentration of WT and KO mice during tick bite, each mouse was challenged by 10 *I. scapularis* nymphs for 48 h. The mice were euthanized by carbon dioxide asphyxiation and cervical dislocation, and then bled by terminal cardiac puncture. The sera from WT ($n = 10$) and KO mice ($n = 10$) were then separated from murine blood samples by centrifugation at $1,000\times$ g for 10 min at 4˚C. The naïve WT ($n = 5$) and KO mice sera ($n = 5$) were also collected. The biopsies of tick bite sites were also collected. The histamine concentration was evaluated by Histamine ELISA kit (Abcam, #ab213975).

## Vascular permeability evaluation

To assess vascular permeability in WT and KO mice during tick bite, the mice were challenged with *I. scapularis* nymphs for 48 h. Then, 50 mg/kg Evans blue (Sigma, #E2129-10G) was injected intravenously into the WT ($n = 6$) and KO mice ($n = 6$). Any extravasated Evans blue present at the tick bite site was extracted in formamide and quantified spectrophotometrically at a wavelength of 610 nm using a Tecan infinite m200 plate reader (Tecan, Switzerland).

## Analysis of immune cells at the tick bite site in WT and KO mice

After feeding by ticks ($n = 10$ per mouse) for 48 h, WT ($n = 4$) and KO mice ($n = 4$) were euthanized by carbon dioxide asphyxiation and cervical dislocation, and one 3.5 mm (Integra Life-Sciences, US) punch biopsy specimen was taken from tick bite site on the ears. The punch biopsy specimens from the same mouse were pooled together for analysis. The biopsy samples were incubated for 1.5 h in Dispase II (Sigma, #SCM133) in DMEM media with 10% FBS and then cut into small pieces. The small pieces were then digested for 1.5 h in collagenase (Gibco, #17100017) in media. The digested samples were then individually passed through 70-μm filters to obtain single-cell suspensions. After washing once with PBS, cells were stained using the LIVE/DEAD fixable violet stain kit (Invitrogen, #L34955). For testing neutrophils, dendritic cells, macrophages, the cells were further incubated with fluorochrome-conjugated monoclonal antibodies against CD45 (PerCP; BD Pharmingen; #561047), CD11b (PE; Biolegend; #101208), CD11c (PE-Cy7; BD Pharmingen; #558079), and Ly6G (FITC; Tonbo; #35–5931) for 30 min at room temperature. For testing mast cells and basophils, the cells were then incubated with FITC anti-mouse CD49b Antibody (BioLegend, #103503), Pacific Blue anti-mouse CD117 (c-Kit) Antibody (Biolegend, #105819), APC anti-mouse CD200R3 Antibody (Biolegend, #142207) for 30 min at room temperature and washed twice with PBS. The samples were run on a BD LSRII flow cytometer and analyzed using FlowJo software. The gating strategy followed our previous studies [40,41].

## Histopathology

Five WT and 4 KO mice with attached ticks were euthanized as described above 96 h post tick placement. The ears ($n = 18$) were removed at the base of the ear and placed in 10% neutral

buffered formalin and submitted to the Comparative Pathology Research (CPR) Core, Histology Laboratory (Department of Comparative Medicine, Yale University School of Medicine) for routine processing into paraffin wax where the ears were subsequently "bread-loafed" into 3 to 5 strips prior to embedding as previously described [42]. The tissue blocks were sectioned at 5 microns and stained for hematoxylin and eosin (HE) by routine methods. The 18 HE-stained slides were evaluated blind to the experimental manipulation (CJB) and semiquantitative analysis as described in Kurokawa, Narasimhan (29). Specifically, each ear was evaluated for the presence or absence of ticks, and the number of bite/inflammatory foci per slide. Each bite/inflammatory foci were scored for edema, hemorrhage, fibrin/necrosis, and inflammation where 5 = severe, 4 = marked, 3 = moderate, 2 = mild, 1 = minimal, scant 0.5, and trace = 0.25. The individual parameters were averaged separately as well as the sum all parameters for each tick bite to give a severity of injury score for each tick bite. The slides were evaluated using an Olympus BX 53 light microscope, photomicrographs taken using an Olympus DP28 camera using Olympus cellSens 4.1 imaging software (Olympus, Evident Scientific, Dallas, Texas, USA), and the images optimized using Adobe Photoshop Creative Cloud 23.0.0 (Adobe, San Jose, California, USA).

## Mouse cytokine/chemokine arrays and qPCR

After 48 h tick bite ($n = 10$ per mouse), WT ($n = 6$) and KO mice ($n = 6$) were euthanized as described above, and one 3.5 mm (Integra LifeSciences, US) punch biopsy specimen was taken from tick bite site on the ears. Total RNA was extracted using RNeasy Fibrous Tissue Mini Kit according to the manufacturer's instructions (Qiagen, # 74704). cDNA was synthesized, and qPCR was performed as described above. In addition, we also quantify cytokines production by the Mouse Cytokine/Chemokine Array 32-plex (MD-32) performed by Eve Technologies. Serum collected from each group of mice was sent for cytokine analyses. The cytokines represented by this array are Eotaxin, G-CSF, GM-CSF, IFN-γ, IL-1α, IL-1β, IL-2, IL-3, IL-4, IL-5, IL-6, IL-7, IL-9, IL-10, IL-12 (p40), IL-12 (p70), IL-13, IL-15, IL-17A, IP-10, KC, LIF, LIX, MCP-1, M-CSF, MIG, MIP-1α, MIP-1β, MIP-2, RANTES, and TNFα.

## Reactive oxygen species (ROS) quantification at the tick bite site

After being fed upon by ticks ($n = 10$ per mouse), WT ($n = 4$) and KO mice ($n = 4$) were euthanized as described above, and one 3.5 mm (Integra LifeSciences, US) punch biopsy specimen was taken from tick bite site on the ears. The tissue was cut off at the base and split into dorsal and ventral halves to get single cells. The samples were added 2′-7′-Dichlorodihydrofluorescein diacetate (DCFDA) from Cellular ROS Assay Kit (Abcam, # ab113851), and then quantified by fluorescence spectroscopy with excitation/emission at 485 nm/535 nm.

## RNA-seq of mice and bioinformatic analyses

WT ($n = 3$) and KO mice ($n = 3$) after 48 h tick bite ($n = 10$ per mouse) were euthanized as described above, and one 3.5 mm (Integra LifeSciences, US) punch biopsy specimen was taken from the tick bite site on the ears. Total RNA was extracted using RNeasy Fibrous Tissue Mini Kit according to the manufacturer's instructions (Qiagen, # 74704). RNA was submitted for library preparation using TruSeq (Illumina, San Diego, California, USA) and sequenced using Illumina HiSeq 2500 by paired-end sequencing at the YCGA. All the RNA-seq analyses were performed using Partek Genomics Flow software (St. Louis, Missouri, USA). Specifically, RNA-seq data were trimmed and aligned to the mouse genome (mm10) using STAR (v2.7.3a) [43]. The aligned reads were quantified to Ensembl Transcripts release 100 using the Partek E/M algorithm [44] and the subsequent steps were performed on gene-level annotation followed

by total count normalization. The gene-level data were normalized by dividing the gene counts by the total number of reads followed by the addition of a small offset (0.0001). Gene expression heatmap were performed using default parameters for the determination of the component number, with all components contributing equally in Partek Flow. Volcano plot was performed on the genes that were differentially expressed across the conditions. KEGG enrichment analysis was conducted using the functional annotation tool DAVID (https://david.ncifcrf.gov/tools.jsp).

## Skin biopsy culture

After 48 h tick bite (*n* = 10 per mouse), WT (*n* = 4) and KO mice (*n* = 4) were euthanized as described above, and the ear skin samples were cleaned well with 70% ethanol. We then performed 3.5 mm (Integra LifeSciences, US) punch biopsies from the tick bite sites of WT and KO mice. The collected skin punches were placed in 1 mL of BSK-H complete medium (Sigma-Aldrich, #B8291) containing rifampicin. Culture tubes were incubated at 33°C and examined every 5 days for the presence of spirochetes by dark-field microscopy and hemocytometer (INCYTO, #DHC-N01) as described previously. The viability of *B. burgdorferi* in the positive tubes were then evaluated using BacTiter-Glo Microbial Cell Viability Assay kit (Promega, #G8230).

## Histamine stimulation and *B. burgdorferi* quantification

To test if histamine affects *B. burgdorferi* quantification by ticks, we intradermally injected 0.3 ug/mL anti-dinitrophenyl (DNP) IgE (Sigma-Aldrich, #D8406-100UG) to stimulate histamine release in KO mice (*n* = 4) according to previous studies [20]. After 24 h, 10 *I. scapularis* nymphs were placed on each *B. burgdorferi*-infected WT (*n* = 4) and KO mouse (*n* = 4) for feeding. The *B. burgdorferi* burden in the tick gut after 48 h feeding was assessed by qPCR as described above.

## Statistical analysis

Statistical significance of differences observed in experimental and control groups was analyzed using GraphPad Prism version 8.0 (GraphPad Software, San Diego, California, USA). Nonparametric Mann–Whitney test or unpaired *t* test were utilized to compare the mean values of control and tested groups, and $p < 0.05$ was considered significant.

## Supporting information

**S1 Fig. No significant difference of ROS level was observed between WT and KO mice at the tick bite site.** Statistical significance was assessed using a nonparametric Mann–Whitney test (ns, $p > 0.05$). Data underlying this figure can be found in S1 Data.
(TIFF)

**S2 Fig. Serum cytokines and chemokines production in WT and KO mice after challenged by *B. burgdorferi*-infected ticks.** Data are represented as mean ± SD. Data underlying this figure can be found in S1 Data.
(TIFF)

**S3 Fig. Gene expression of cytokines and chemokines at the tick bite of WT and KO mice after challenged by *B. burgdorferi*-infected ticks.** Data are represented as mean ± SD. Data underlying this figure can be found in S1 Data.
(TIFF)

**S4 Fig. Adiponectin or associated factors influence *B. burgdorferi* survival at the tick bite site.** (A) *B. burgdorferi* in the tick bite biopsy from KO mice has higher viability than in WT mice as determined by BacTiter-Glo assay. (B) No significant difference of *B. burgdorferi* burden in ticks feeding on WT and KO mice after stimulating histamine release in KO mice. For all the data, statistical significance was assessed using a nonparametric Mann–Whitney test (*$p < 0.05$). Data underlying this figure can be found in S1 Data.
(TIFF)

**S1 Table. The differently expressed genes of transcriptome of ticks feeding on adiponectin WT and KO mice.**
(DOCX)

**S2 Table. The differently expressed genes of transcriptome of WT and KO murine skin during tick bite.**
(DOCX)

**S3 Table. The primers used in this study.**
(DOCX)

**S1 Data. An excel file with the detailed individual numerical values that underlie the summary data displayed in the figure panels: Figs 1A, 1C–1E, 2A, 2B, 2D–2F, 3A–3D, 4A–4E, S1–S4.**
(XLSX)

# Acknowledgments

We sincerely thank Dr. Sukanya Narasimhan, Dr. Jesse Hwang, Dr. Sameet Mehta, Dr. Yu-Min Chuang, Ms. Kathleen DePonte, and Mr. Ming-Jie Wu for their excellent technical assistance.

# Author Contributions

**Conceptualization:** Xiaotian Tang, Erol Fikrig.

**Data curation:** Xiaotian Tang, Yongguo Cao, Carmen J. Booth, Gunjan Arora, Yingjun Cui, Jaqueline Matias.

**Formal analysis:** Xiaotian Tang, Yongguo Cao, Carmen J. Booth, Gunjan Arora, Yingjun Cui, Jaqueline Matias.

**Funding acquisition:** Erol Fikrig.

**Investigation:** Xiaotian Tang, Yongguo Cao, Gunjan Arora, Yingjun Cui, Jaqueline Matias, Erol Fikrig.

**Methodology:** Xiaotian Tang, Yongguo Cao, Carmen J. Booth, Gunjan Arora, Yingjun Cui, Jaqueline Matias.

**Project administration:** Erol Fikrig.

**Resources:** Erol Fikrig.

**Software:** Xiaotian Tang, Carmen J. Booth.

**Supervision:** Erol Fikrig.

**Validation:** Xiaotian Tang, Erol Fikrig.

**Visualization:** Xiaotian Tang, Carmen J. Booth.

**Writing – original draft:** Xiaotian Tang, Carmen J. Booth, Erol Fikrig.

**Writing – review & editing:** Xiaotian Tang, Yongguo Cao, Carmen J. Booth, Gunjan Arora, Yingjun Cui, Jaqueline Matias, Erol Fikrig.

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
