## [Editor Report · Decision Letter 0]

17 May 2023

Dear Dr. Tang, 

Thank you for submitting your manuscript entitled "Adiponectin inhibits acquisition of the Lyme disease agent by ticks" for consideration as a Short Reports by PLOS Biology.

Your manuscript has now been evaluated by the PLOS Biology editorial staff and I am writing to let you know that we would like to send your submission out for external peer review.

Once your full submission is complete, your paper will undergo a series of checks in preparation for peer review. After your manuscript has passed the checks it will be sent out for review. To provide the metadata for your submission, please Login to Editorial Manager (https://www.editorialmanager.com/pbiology) within two working days, i.e. by May 19 2023 11:59PM.

Kind regards,

Paula

---

Senior Editor

PLOS Biology

---

## [Decision Letter · Decision Letter 1]

23 Jun 2023

Dear Dr. Tang,

Thank you for your patience while your manuscript "Adiponectin inhibits acquisition of the Lyme disease agent by ticks" was peer-reviewed at PLOS Biology. It has now been evaluated by the PLOS Biology editors, an Academic Editor with relevant expertise, and by several independent reviewers. 

In light of the reviews, which you will find at the end of this email along with the comments from the Academic Editor, we would like to invite you to revise the work to thoroughly address the reviewers' reports.

As you will see below, the reviewers have important issues that should be solved before further consideration. In particular, the reviewers think that some of your conclusions are overstatements, the manuscript is in need of clarifications and in need of adding information to the methods. Regarding this last point, it is very important that you add the number of mice, bacteria and ticks, in particular the bacterial burden in the in vivo models. Reviewer #1 finds the histopathology data inadequate. All those issues will need to be solved before further consideration of the manuscript.

Given the extent of revision needed, we cannot make a decision about publication until we have seen the revised manuscript and your response to the reviewers' comments. Your revised manuscript is likely to be sent for further evaluation by all or a subset of the reviewers.

**IMPORTANT - SUBMITTING YOUR REVISION**

*Re-submission Checklist*

*Published Peer Review*

*PLOS Data Policy*

*Blot and Gel Data Policy*

Sincerely,

Paula

---

Senior Editor

PLOS Biology

REVIEWS:

Reviewer #1: Tick-borne bacterial pathogens.

Reviewer #2: Host-pathogen interactions.

Reviewer #3: Lyme disease.

Reviewer #1: The authors make the argument that adiponectin, an abundant adipocyte-derived hormone in humans, inhibits Ixodes scapularis acquisition of Borrelia burgdorferi, the agent of Lyme disease. The authors first demonstrate that B. burgdorferi levels are increased in ticks fed on adiponectin deficient mice compared to wild type mice and attribute elevated mRNA for three tick genes to adiponectin. The investigators then compare various aspects of the tick feeding site between C57BL/6 adiponectic deficient and C57BL/6 wild type mice. The data suggest that adiponectic induces host histamine release in response to tick feeding, which in turn leads to vascular leakage. The adiponectin deficient mice also have reduced numbers of neutrophils, macrophages, and inflammation at the bite site as well as reduced IL-12 and IL-1b. The possible effect of adiponectin on tick acquisition of B. burdorferi is a potentially important finding.

With a few exceptions, the conclusion are supported by the data. The authors do overstate causation when correlation would be more appropriate. Specific examples are listed below. The histopathology data is inadequate. The methods require some additional details and clarification.

Line 41 Abstract: The authors state that, "All these factors mediated by adiponectin influence B.

burgdorferi survival at the interface of tick bite site. This study represents a conceptual

advancement that a host adipocyte-derived hormone modulates pathogen acquisition by

a blood feeding arthropod." These conclusions are somewhat overstated in that the direct effect of adiponectin is not truly measured nor is survival of the B. burgdorferi at the tick bite site.

Line 103: The authors state that B. burgdorferi levels were equivalent in the WT and adiponectin deficient mice. The number of mice used in this experiment should be reported. Additionally, the actual quantity of B. burgdorferi in the mice is essential for interpreting the difference in B. burgdorferi levels in tick midguts and thus should be reported. 

Line 121: The authors state that, "We examined whether incoming blood adiponectin influences tick physiology or alters the local host environment at the tick bite site." The authors do not demonstrate that there is measurable adiponectin in the blood meal of the wildtype mice. Thus, the changes in tick gene expression may be due to other factors associated with adiponectin deficiency rather than directly due to differences in adiponectin levels. This conclusion needs to be more accurately stated.

Line 238: The authors state, "In our study, higher levels of histamine in WT mice increased vascular leakage, which led to the additional recruitment of neutrophils and macrophages to the tick bite site

and caused severe inflammation." This is an over statement of causation. There are many other factors besides vascular permeability that can lead to recruitment of neutrophils and macrophages. This conclusion should be more accurately stated and the other possible causes of increased neutrophils and macrophage at the bite site should be addressed. 

Line 254. The authors state, "In summary, we have discovered that host serum adiponectin affects the initial entry of B. burgdorferi into ticks." This is an overinterpretation of the data. Please edit appropriately. 

Methods and Results: Number of mice used in each experiment should be stated.

Line 290: Please clarify how qPCR analysis was done to enumerate B. burgdorferi levels in the tick midgut. Presumably this was done using genomic DNA, please clarify. It seems unusual to report ng of nucleic acid when the readout from qPCR is either copy number or relative expression (if using mRNA). Please describe how the data analysis was done to determine ng of amplicon was determined.

Fig 1A - should indicate that the mice were needle inoculated with B. burgdorferi.

Line 288: The oligonucleotides for PCR should be included. 

Line 311: The title for the section "Splenocyte viability evaluation" requires editing for accuracy or clarity.

Line 359: Was the tick bite site or the ears used for immune cell analysis? Please clarify.

Line 380: Please provide more details about the histopathology scoring system. How were sections selected for evaluation? How many sections were evaluated? Please clarify the criteria for minimal, mild, moderate, marked and severe? Presumably this is inflammation, but this should be stated. Please include representative images or remove these data.

Reviewer #2: The manuscript "Adiponectin inhibits acquisition of Lyme disease agent by ticks" was a pleasure to read and presents potentially impactful findings. Further, many findings of this paper open more questions ripe for future investigation. As interesting as I find the study, I would like a few points addressed. 

The Authors report that ticks fed on infected adipo -/- mice had significantly elevated Borrelia acquisition compared to feeding on WT mice. However, this did not impact the feeding success of ticks. 

* One key detail that is stated, but not shown (Line 101-102 Figure A), is the level of Borrelia in the mice. Figure A shows a diagram of the experiment, but the bacterial burden in the mice is missing. The bacterial burden from these mice groups should be included to show the acquisition difference is not reflecting the burden in the mice. 

* Numbers of animals need to be stated throughout for the reader to judge rigor of each experiment

o Please include the numbers of ticks and mice used in Fig1B,C,D. Are these all the same mice, or are different mice in A,B,C. If they are stating the engorgement weight does not differ, I would like to see weights of the ticks used in FigB. Just from counting dots on the graph, it appears there are more ticks in Fig 1D. 

o Are the mice used in Fig1 C, D infected?

o From the Fig1B graph it appears that only 12 ticks are in the adipo-/- group. How many mice are these are these ticks from, how many ticks per mouse fed?

To investigate what may be causing the increased Borrelia acquisition in the adiponectin -/- fed ticks they performed RNAseq on ticks collected from wt and adipo -/- mice. From this data, they found histamine binding proteins were differentially expressed between wt and adiponectin -/- fed ticks. They found that adiponectin -/- fed ticks had lower levels transcription of HBPs. They did not explore the downstream impacts of these HBPs in the tick, but instead it prompted them to investigate the levels of histamine between WT and adipo-/- mice. 

* A strength, but also a weakness, in their RNA seq data is that it was done in the absence of Borrelia infection. This does isolate the variable of adipo-/- better but also lessens the direct relevance to Borrelia acquisition. I would have liked to see an experiment showing the HBP expression differences hold in ticks fed on infected adipo-/- vs wt mice. If not part of RNAseq study, could transcription of the HBP genes be measured by qPCR from the tick RNA samples used in figure 1B,1D?

The differences in HBP expression prompted them to investigate the levels of histamine, and its downstream consequences at the tick bite site. They showed that the adipo-/- mice secreted less histamine, resulting in less immune cell infiltration and attenuated inflammation at the bite site. These results demonstrate that the adipo-/- mice here a greatly dampened immune reaction at the bite site, which is consistent with the the increased Borrelia acquisition by ticks from the adipo-/- ticks.

* These findings are consistent with the higher Borrelia acquisition by ticks fed on adipo-/- mice, however they have not directly tested the impact of adiponectin dependent histamine levels on Borrelia acquisition. Could a rescue experiment address this? Could infected adipo-/- mice be supplemented to stimulate histamine release, and/or could WT mice be treated with anti-histamines, to see it is reduces the differences in Borrelia acquisition. Alternatively, if the effect of histamine on Borrelia acquisition was directly tested previously, an expanded discussion could address this concern.

Reviewer #3: I have only a few suggested revisions/questions regarding this innovative and well-designed study.

1. Line 72-74. Why would a high concentration of adiponectin per se cause varied functions in multiple organs? Might moderate or even low concentrations cause varied functions in multiple organs?

2. Line 102. It would be useful to mention here how experimental and control mice were infected that ensured a similar B. burgdorferi burden even though the infection protocol is described in the Methods section of the paper.

3. Line 111-112. How does the hormone adiponectin encode a C1q domain? 

4. Lines 248-249. Are adiponectin levels affected by the amount of B. burgdorferi introduced by tick bite and/or the number tick bite exposures in the mammalian host?

5. Line 274-276. How does the injection of 1 x 105 cells/ml B. burgdorferi compare with the amount of B. burgdorferi introduced during I. scapularis nymphal feeding and does it matter in regard to experimental outcomes?

6. Lines 379-381. Does this sentence describe the inflammation severity score? If so, was a more objective quantitative unstated definition used to assess the score (e.g., number inflammatory cells for each severity category)?

7. Figure 3B. The pink arrows mentioned in the Figure legend are difficult to visualize in Figure 3B.

Minor

8. Line 114. Substitute "directly" for "direct".

9. Line 143. Substitute "levels" for "level".

10. Line 219. Suggest revising the sentence to read: "In this study, we found that host derived adiponectin influences B. burgdorferi acquisition by ticks, which is the first evidence that a host factor in blood modulates pathogen acquisition by arthropod vector.

11. Line 258. Substitute " modulate" for "modulates"

12. Line 396. Substitute "Reactive oxygen species (ROS) quantification…" for "ROS quantification…"

13. Line 611. Substitute "…at the tick bite site of WT…" for "at the tick bite of WT…"

14. Line 572 (Figure 1A legend). Substitute "(A) Pathogen-free I. scapularis nymphs 

were fed on… for "(A) Ticks were fed on…"

COMMENTS FROM THE ACADEMIC EDITOR:

I cannot judge the need for additional experiments regarding histopathology because this is outside my area of technical expertise. However I do agree with several key points raised by Reviewer 1, which overlap with some of the comments made by the other two reviewers.

- First, there is a general issue of overinterpretation. The conclusion that adiponectin influences Borrelia acquisition by ticks is overstated in that the direct effect of adiponectin is not truly measured. The difference between WT and adiponectin-deficient mice could be due to other factors associated with adiponectin deficiency rather than directly due to differences in adiponectin levels in the blood meal. Also, the claim that "adiponectin influences B. burgdorferi acquisition by ticks, which is the first evidence that a host factor in blood modulates pathogen acquisition by arthropod vector" (lines 219-221) is untrue. The authors themselves explain in the introduction that host blood metabolites can directly or indirectly influence arthropod susceptibility to pathogens. They cite the example of the host-derived cytokine in blood IFN-γ, which interacts with tick receptor Dome1 and activates the STAT-dependent pathway, limiting B. burgdorferi persistence in Ixodes scapularis.

- Second, a critical piece of information is missing. The authors state that WT and adiponectin-deficient mice had "a similar B. burgdorferi burden" (line 102) but they did not show the data supporting this statement. This information is essential and must be duly justified. It should include a measure of Borrelia survival at the tick bite site.

---

## [Editor Report · Decision Letter 2]

4 Sep 2023

Dear Dr Tang,

Thank you for your patience while we considered your revised manuscript "Adiponectin deficiency alters acquisition of the Lyme disease agent by ticks" for publication as a Short Reports at PLOS Biology. This revised version of your manuscript has been evaluated by the PLOS Biology editors and the Academic Editor.

Based on our Academic Editor's assessment of your revision, we are likely to accept this manuscript for publication, provided you satisfactorily address the following data and other policy-related requests.

1. ETHICS STATEMENT:

-- Please include the full name of the IACUC/ethics committee that reviewed and approved the animal care and use protocol/permit/project license. Please also include an approval number.

-- Please include the specific national or international regulations/guidelines to which your animal care and use protocol adhered. Please note that institutional or accreditation organization guidelines (such as AAALAC) do not meet this requirement.

2. DATA POLICY:

A) Supplementary files (e.g., excel). Please ensure that all data files are uploaded as 'Supporting Information' and are invariably referred to (in the manuscript, figure legends, and the Description field when uploading your files) using the following format verbatim: S1 Data, S2 Data, etc. Multiple panels of a single or even several figures can be included as multiple sheets in one excel file that is saved using exactly the following convention: S1_Data.xlsx (using an underscore).

B) Deposition in a publicly available repository. Please also provide the accession code or a reviewer link so that we may view your data before publication.

Regardless of the method selected, please ensure that you provide the individual numerical values that underlie the summary data displayed in the following figure panels as they are essential for readers to assess your analysis and to reproduce it: Figures 1ACDE, 2ABDEF, 3ABCD, 4ABCDE, and Supplementary Figures S1, S2, S3, S4AB.

**NOTE: the numerical data provided should include all replicates AND the way in which the plotted mean and errors were derived (it should not present only the mean/average values).**

**Please also ensure that figure legends in your manuscript include information on where the underlying data can be found, and ensure your supplemental data file/s has a legend.**

3. Please provide a blurb which (if accepted) will be included in our weekly and monthly Electronic Table of Contents, sent out to readers of PLOS Biology, and may be used to promote your article in social media. The blurb should be about 30-40 words long and is subject to editorial changes. It should, without exaggeration, entice people to read your manuscript. It should not be redundant with the title and should not contain acronyms or abbreviations.

4. We suggest a change in the title to "Adiponectin in the mammalian host influences ticks' acquisition of the Lyme disease pathogen Borrelia".

We expect to receive your revised manuscript within two weeks.

*Published Peer Review History*

*Press*

Sincerely,

Paula

---

Senior Editor,

pjaureguionieva@plos.org,

PLOS Biology

---

## [Editor Report · Decision Letter 3]

12 Sep 2023

Dear Dr Tang,

Thank you for the submission of your revised Short Reports "Adiponectin in the mammalian host influences ticks' acquisition of the Lyme disease pathogen Borrelia" for publication in PLOS Biology. On behalf of my colleagues and the Academic Editor, Louis Lambrechts, I am pleased to say that we can in principle accept your manuscript for publication, provided you address any remaining formatting and reporting issues. These will be detailed in an email you should receive within 2-3 business days from our colleagues in the journal operations team; no action is required from you until then. Please note that we will not be able to formally accept your manuscript and schedule it for publication until you have completed any requested changes.

PRESS

Sincerely, 

Paula 

---

Senior Editor

PLOS Biology
